# Radiographic and Clinical Assessment of Unidirectional Porous Beta-Tricalcium Phosphate to Treat Benign Bone Tumors

**DOI:** 10.3390/biomimetics10020101

**Published:** 2025-02-10

**Authors:** Toshiyuki Kunisada, Eiji Nakata, Tomohiro Fujiwara, Haruyoshi Katayama, Takuto Itano, Takanao Kurozumi, Teruhiko Ando, Toshifumi Ozaki

**Affiliations:** 1Department of Orthopaedic Surgery, Okayama University Graduate School of Medicine, Dentistry, and Pharmaceutical Sciences, 2-5-1, Shikata-cho, Okayama 700-8558, Japan; 2Department of Medical Materials for Musculoskeletal Reconstruction, Okayama University Graduate School of Medicine, Dentistry, and Pharmaceutical Sciences, 2-5-1, Shikata-cho, Okayama 700-8558, Japan

**Keywords:** unidirectional porous beta-tricalcium phosphate, bone tumor, bone graft, radiography, bone remodeling

## Abstract

The purpose of this study was to evaluate radiographic changes, clinical outcomes, and complications following unidirectional porous beta-tricalcium phosphate (UDPTCP) implantation for the treatment of benign bone tumors. We retrospectively analyzed 46 patients who underwent intralesional resection. The patients were divided into two cohorts: Cohort 1 (n = 32), which included all bones except the phalanges and metacarpal/tarsal bones, and Cohort 2 (n = 14), which included the phalanges and metacarpal/tarsal bones. Radiographic changes were assessed at each reading based on resorption of the implanted UDPTCP and bone trabeculation through the defect. UDPTCP resorption and bone trabeculation were observed on radiographs within 3 months of surgery in all patients. Bone remodeling in the cavity progressed steadily for up to 3 years postoperatively. In Cohort 1, resorption and trabeculation progressed significantly in young patients, and trabeculation developed significantly in small lesions. The rates of resorption and trabeculation at 3 months postoperatively correlated statistically with their increased rates at one year. There was no statistical difference in resorption and trabeculation rates between Cohort 1 and Cohort 2. There were no cases of postoperative deep infections or allergic reactions related to the implant. UDPTCP is a useful bone-filling substitute for the treatment of benign bone tumors and has a low complication rate.

## 1. Introduction

Benign bone tumors frequently necessitate surgical intervention when they induce significant symptoms or pain, or when they elevate the risk of pathological fractures. The standard surgical procedure for treating this condition is intralesional resection (curettage). Tumor resection results in the formation of severe bone defects, necessitating the filling of the resulting cavity to restore mechanical strength and prevent pathological fractures. Autogenous bone grafting has long been regarded as the gold standard. Nevertheless, complications have been documented, including donor-site fracture and infection, prolonged surgical time, increased blood loss, and limited bone supply [1]. A variety of artificial bone graft substitutes with diverse compositions, porous structures, and porosities have been developed and employed in the context of bone tumor surgery [2,3]. Beta-tricalcium phosphate (β-TCP) is one of the most commonly utilized synthetic bone graft substitutes, exhibiting osteoconductive properties and integration into the surrounding bone [4]. However, implanted β-TCP may occasionally undergo degradation, particularly in weight-bearing regions, due to its inherent mechanical deficiencies stemming from its high porosity [5].

Unidirectional porous beta-tricalcium phosphate (UDPTCP; Affinos^®^, Kuraray Co., Ltd., Tokyo, Japan) was developed with an interconnected porous structure comprising unidirectional oval pores in the horizontal direction (approximately 100–300 µm in the longest diameter) that are fully penetrating throughout the material [6]. The initial compression strengths of 8 and 1.5 MPa were applied in the directions parallel and perpendicular to the pores, respectively. Several experiments have been conducted on UDPTCP implantation in animals, and histological evaluation has revealed the presence of early bone formation and new angiogenesis, which align with the porous direction throughout the material [6,7,8]. Clinically, despite the limited number of available papers, UDPTCP has been used to fill gaps in various surgical procedures, including fracture repair, osteotomy, and spinal fusion. These studies have demonstrated that UDPTCP has acceptable clinical performance as a bone graft substitute in orthopedic surgery [9,10,11]. A limited number of studies have presented radiographic evaluations following UDPTCP implantation in the context of bone tumor surgeries [7,12,13]. However, the studies in this area have only presented case series or outcomes within a limited postoperative timeframe.

The objective of this study was to analyze radiographic changes, clinical outcomes, and complications, particularly those occurring more than one year postoperatively, in patients who underwent UDPTCP implantation following surgical resection of bone tumors. Additionally, the study aimed to discuss the clinical benefits of UDPTCP implantation in bone tumor surgeries.

## 2. Materials and Methods

A retrospective analysis was conducted on 46 patients who had undergone intralesional resection (curettage) and UDPTCP implantation for the treatment of benign bone tumors between the years 2015 and 2021. Patients who had undergone surgical treatment for local recurrence were excluded from the study. Intralesional resection was conducted using manual curettage alone for less aggressive benign bone tumors. Intralesional resection was conducted using a high-speed burr for the treatment of aggressive benign bone tumors, including giant cell tumor (GCT), aneurysmal bone cyst (ABC), and chondroblastoma (CB). Adjuvant therapies for GCT included the use of an argon beam coagulator, soaking with alcohol, or irrigation with distilled water. The mean follow-up period was 37 months (range, 12–84 months).

Adequate amounts of both block- and granule-type UDPTCP were implanted into bone defects after intralesional resection of the tumor in 24 patients. In the remaining cases, the treatment regimen consisted of granule-type UDPTCP in 20 patients and block-type UDPTCP in 2 patients. Bone defects with a longitudinal diameter of 3 cm or less were typically filled with either granule-type UDPTCP or block-type UDPTCP alone. The block-type UDPTCP was grafted into the cavity with its pores aligned parallel to the longitudinal axis of the bone, as previously described [14]. In 3 patients, autogenous bone was simultaneously grafted into subchondral bone defects adjacent to the joint. Four patients underwent internal fixation with plates and screws to repair a pathological fracture that had occurred at the time of diagnosis or to prevent postoperative fracture and facilitate rehabilitation. The occurrence of any clinical complications was determined through a review of the medical records and was assessed by the attending surgeon (TK). Patients were divided into two cohorts according to Ikuta’s classification [12], as the process of bone regeneration may differ due to the significantly smaller lesions observed in the fingers and toes. Cohort 1 comprised all bones other than phalanges and metacarpal/tarsal bones, while Cohort 2 consisted of phalanges and metacarpal/tarsal bones.

Subsequent to the surgical procedure, radiographic images were obtained at the outpatient clinic. The volume of the bone tumor was calculated by multiplying the anteroposterior, transverse, and cephalocaudal dimensions of the lesion, as measured on the preoperative radiographs. Two physicians in our department (EN and TF), who were blinded to the clinical conditions, evaluated radiographic changes in the implanted UDPTCP according to Anker’s classification [15]. The resorption of the implanted UDPTCP and bone trabeculation through the defect were divided into five stages, with each stage rated as 0%, 25%, 50%, 75%, or 100%.

Univariate analyses were conducted using the Mann–Whitney U test and the Kruskal–Wallis test for non-parametric data. For all tests, a statistical significance level of *p* < 0.05 was employed. All analyses were conducted using the software program STATA/SE 18.0 for Mac (StataCorp, College Station, TX, USA).

## 3. Results

### 3.1. Clinical Outcomes

The study population consisted of 21 males and 25 females with an average age of 25.6 years (range, 6–70 years). The histology included 14 enchondroma (EC), 12 simple bone cyst (SBC), 8 CB, 4 fibrous dysplasia (FD), 3 GCT, 3 ABC, 1 osteoid osteoma, and 1 intraosseous ganglion. A total of 32 tumors were classified into Cohort 1 and 25 tumors were located in the lower extremities (9 femur, 9 calcaneus, 5 tibia, 1 patella, and 1 talus), 6 in the upper extremities (4 humerus, 1 radius, and 1 ulna), and 1 in the pelvis (Table 1). A total of 14 tumors were classified into Cohort 2 and 8 tumors were located in the phalanx, 5 in the metacarpal, and 1 in the metatarsal (Table 2). The mean tumor volume was 14.4 cm^3^ (range, 0.94–50.4 cm^3^) in Cohort 1 and 1.2 cm^3^ (range, 0.2–2.6 cm^3^) in Cohort 2. The mean volume of the implanted UDPTCP was 9.5 g (range, 2–25 g) in Cohort 1 and 2.6 g (range, 2–5 g) in Cohort 2.

One case of osteoid osteoma exhibited delayed wound healing, which may have been attributable to a minor superficial infection. Complete healing was achieved 4 weeks after surgery, with only daily wound management required. No cases of postoperative deep infections or allergic reactions related to the implanted UDPTCP were identified. One patient with SBC and a preoperative fracture experienced local recurrence 1 year postoperatively. The patient was monitored at the outpatient clinic and exhibited no further progression of the recurrence nor any impairment of daily activities 5 years postoperatively.

### 3.2. Radiographic Outcomes

The radiographic examination revealed the resorption of the implanted UDPTCP and bone trabeculation through the defect in all patients within 3 months after surgery (Figure 1). Three months after surgery, the mean resorption rates of the implanted UDPTCP were 36.3% in Cohort 1 and 34.1% in Cohort 2, while the mean rates of bone trabeculation through the defect were 19.4% in Cohort 1 and 18.2% in Cohort 2. Twelve months after surgery, the mean resorption rate was 82.8% in Cohort 1 and 89.3% in Cohort 2, while the mean rate of bone trabeculation was 73.4% in Cohort 1 and 82.1% in Cohort 2. The rates of implant resorption and bone trabeculation exhibited a notable increase within the first year postoperatively in all lesions of Cohorts 1 and 2 (Figure 2 and Figure 3). The rates demonstrated a continuous increase in both cohorts at 1 year postoperatively. By 3 years postoperatively, the rates reached a plateau. A total of 75% UDPTCP resorption was observed in all patients, with an average of 11.2 months (range, 3–36 months) postoperatively in Cohort 1 and 9.6 months (range, 3–18 months) in Cohort 2. One hundred percent resorption was identified in 25 patients (78%) in Cohort 1 and all patients in Cohort 2, with an average of 16.1 months (range, 3–48 months) postoperatively and 14.6 months (range, 6–36 months), respectively. Furthermore, 75% trabeculation was observed in all patients, with an average of 14.6 months (range, 6–48 months) postoperatively and 13.3 months (range, 6–48 months), respectively. One hundred percent trabeculation was identified in 20 patients (63%) in Cohort 1 and 13 patients (93%) in Cohort 2, with an average of 18.9 months (range, 6–48 months) postoperatively and 14.8 months (range, 6–24 months), respectively. The rates of resorption and trabeculation in Cohort 1 were observed to be lower than those in Cohort 2 on the radiographs from 3 months postoperatively onwards. However, no significant difference was identified between the rates of the two cohorts.

Table 3 illustrates the correlation between radiographic assessment and clinical factors in Cohort 1. The Mann–Whitney U test demonstrated that age ≤15 years was a statistically significant factor associated with UDPTCP resorption and bone trabeculation of the defect in Cohort 1. Additionally, tumor size was identified as a statistically significant predictor of bone trabeculation in Cohort 1. An increased resorption rate of the implant was observed in patients with small tumors, although this difference was not statistically significant. No differences were observed in implant resorption and bone trabeculation at the site in the long bone. Table 4 illustrates the correlation between radiographic assessment and clinical factors in Cohort 2. Although not statistically significant, younger patients and smaller tumor sizes were observed to demonstrate an increased rate of implant resorption and bone trabeculation in Cohort 2.

In Cohort 1, the Kruskal–Wallis test revealed a significant correlation between the rates of implant resorption and bone trabeculation 3 months after surgery and their increased rates 1 year after surgery (*p* < 0.05). Furthermore, the implant resorption rate 1 month after surgery was also found to be significantly associated with an increased rate 1 year after surgery (*p* < 0.05). In Cohort 2, no correlation was observed between the radiographic assessment conducted 1 year after surgery and that conducted earlier.

## 4. Discussion

The residual cavity resulting from the resection of benign bone tumors can occasionally be considerable in size. It is, therefore, essential to fill the cavity in order to restore the mechanical strength of the bone. UDPTCP exhibits a distinctive interconnected porous structure comprising unidirectional oval pores in the horizontal direction, which are fully integrated into the material. The pore size (approximately 100–300 μm in the longest diameter) and microstructure can facilitate the invasion of cells and fluids necessary for osteogenesis. As a result of these characteristics, histological assessment demonstrated the formation of new bone throughout the material 6 weeks following implantation [16]. The replacement of host bone was observed in the cortical bone region, while resorption and remodeling of the material were observed to progress in the medullary cavity. In clinical practice, the use of UDPTCP for the treatment of various bone defects has also yielded positive outcomes, as evidenced by reports in the literature [11,13,17]. This study employed UDPTCP as a bone grafting material in the treatment of benign bone tumors, and its radiographic and clinical outcomes were subsequently evaluated. The radiographic observations were consistent with the histological findings previously reported, and UDPTCP implantation resulted in favorable bone remodeling in defects with minimal complications following the intralesional resection of benign bone tumors.

In the present study, the resorption of implanted UDPTCP and bone trabeculation were identified on radiographs in all patients as early as 1 month after grafting into the defect when treating benign bone tumors. These findings demonstrate that UDPTCP can stimulate early and reliable bone formation, which is a significant advancement in the field of bone tumor treatment. Histological assessments in rabbits revealed the formation of new bone throughout the interior of UDPTCP materials [6], as well as the development of new angiogenesis and blood flow along with new bone formation within the materials 6 weeks after implantation into the tibial defect [8]. These findings suggest that UDPTCP may serve as an optimal material for the restoration of bone defects through early bone formation. A previous study examined the healing process of bone defects filled with UDPTCP within 1 year after surgery for benign bone lesions and reported that the values of material resorption and trabeculation increased steadily with time [12]. The current study corroborates the previous findings of optimal bone remodeling in all lesions across both cohorts. Furthermore, the radiographic changes in the implanted UDPTCP were evaluated over an extended period. In the majority of cases, complete resorption of the UDPTCP and trabeculation within the defect were evident approximately 1.5 years postoperatively. UDPTC resorption and bone trabeculation exhibited continuous progression from 1 year postoperatively, reaching a plateau at 3 years postoperatively. The remodeling of bone in the cavity with an implanted UDPTCP may continue at a steady pace for up to 3 years following the resection of bone tumors.

A number of studies have demonstrated the clinical application of UDPTCP in orthopedic surgeries. The early resorption and replacement of UDPTCP by the host bone were identified in the treatment of distal radial fractures with UDPTCP implantation to fill a bone defect [10,17]. The application of UDPTCP filling led to the restoration and preservation of the anatomic position subsequent to the correction of distal radius fractures. The range of motion of the wrist was restored to its normal range, and the clinical results at the final follow-up were excellent. Imaging assessment confirmed similar findings of progressive replacement of UDPTCP with the host bone in the filling of gaps in femoral fractures [7] and open-wedge high tibial osteotomies [11]. The remodeling of UDPTCP implants used for high tibial osteotomy was detected by computed tomography (CT) at an earlier stage than that of TCP with a conventional spherical interconnected porous structure [11]. The present study demonstrated the occurrence of early bone remodeling and the attainment of favorable clinical outcomes without the occurrence of significant complications following the grafting of UDPTCP for the treatment of benign bone tumors. These clinical findings indicate that UDPTCP exhibits excellent clinical performance as a bone graft substitute in orthopedic surgery.

In Cohort 1, there was a notable advancement in UDPTCP resorption and bone trabeculation among the younger patients. In their regression model for assessing the radiographic change of implanted UDPTCP for benign bone tumors, Ikuta et al. [12] did not report age as an independent factor. In other clinical reports, the impact of younger age on bone remodeling in implanted UDPTCP has not been extensively explored [7,10,11,17]. The previous studies primarily focused on radiographic alterations in middle-aged and elderly patients. The current study included a relatively higher proportion of pediatric patients when compared to previous studies. A study on unidirectional porous hydroxyapatite, a distinct material with analogous structural characteristics, similarly revealed that implant resorption and bone marrow remodeling were markedly more prevalent in pediatric patients with benign bone tumors [14]. Patients of a younger age, particularly those in the pediatric demographic, tend to exhibit a more active process of bone remodeling than individuals of middle age or advanced age. This may provide an explanation as to why age was identified as a significant indicator in Cohort 1 of the present study. However, young age was not identified as an independent factor influencing bone remodeling in Cohort 2. This may be due, in part, to the fact that the histology observed in Cohort 2 was exclusively enchondroma, which typically arises in middle-aged patients. Cohort 2 included a smaller number of pediatric patients and exhibited a higher mean age than Cohort 1.

It is important to note that the results of this study are subject to several significant limitations. Firstly, there were no comparison groups and no randomization. The present study describes patients who underwent intralesional resection followed by UDPTCP implantation. As all patients received the same treatment, a comparison with other types of bone-filling materials or no filling could not be performed. Given the considerable variations in size, location, cortical thickness, and bone tumor histology, it was not feasible to establish a control group for comparison. Secondly, given that benign bone tumors typically manifest in younger individuals, the majority of patients in our series were relatively young, with an average age of 25.6 years. This may have resulted in enhanced bone formation compared to elderly patients. While our results may not support the use of UDPTCP implantation in the older population, previous studies have reported positive clinical outcomes in the treatment of degenerative diseases such as osteoarthritis [11] and cervical myelopathy [16]. It is our contention that UDPTCP can serve as a reliable bone graft substitute in elderly patients. Thirdly, a three-dimensional imaging assessment was not conducted. While CT may be able to more accurately demonstrate the incorporation of implanted UDPTCP and new bone formation for research purposes, repeated CT scans are not necessary for routine clinic after this type of surgy and may not be ethically justifiable due to the issue of radiation exposure, especially in young patients. We contend that plain radiography is an efficacious method for evaluating such changes.

## 5. Conclusions

The radiographic examination revealed the early and excellent resorption of the implanted UDPTCP and bone trabeculation following the resection of the bone tumor and the filling of the bone defect with UDPTCP. Bone remodeling in the cavity with UDPTCP exhibited consistent progression over a period of up to 3 years following the resection of bone tumors. The results of this study demonstrate that UDPTCP is an effective bone-filling substitute for the treatment of benign bone tumors, with a low complication rate.

## Figures and Tables

**Figure 1 biomimetics-10-00101-f001:**
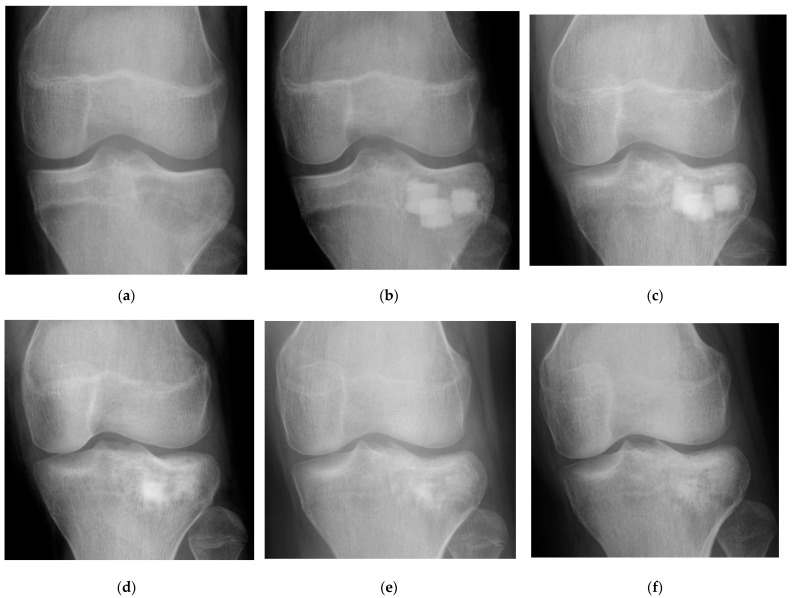
A 13-year-old female patient was treated for a chondroblastoma with grafting of a block-type UDPTCP. Autogenous bone collected from the iliac bone was simultaneously grafted into subchondral bone defects adjacent to the joint. (**a**) Preoperative radiograph image showing a bone defect in the proximal tibia. (**b**) A clear UDPTCP margin was observed 1 week postoperatively in the anteroposterior view of the radiograph. (**c**) Mild implant resorption (25%) and bone trabeculation through the defect (25%) were identified on radiography 3 months after surgery. (**d**) Good implant resorption (75%) and bone trabeculation (75%) were observed 6 months after surgery. (**e**) Almost complete resorption (100%) and bone trabeculation (100%) were observed 1 year after surgery. (**f**) No local tumor recurrence was observed 3 years after surgery, with complete resorption (100%) and trabeculation (100%).

**Figure 2 biomimetics-10-00101-f002:**
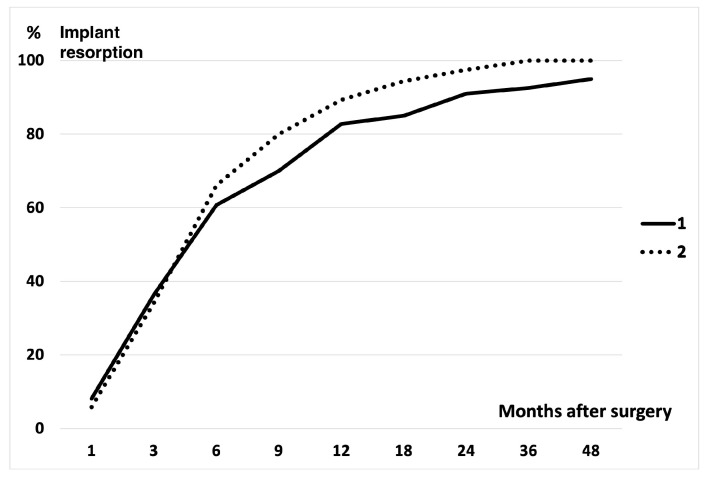
The resorption of implanted UDPTCP over time is shown graphically in the comparison between cohorts 1 and 2. The rate of implant resorption increased significantly within 1 year postoperatively in cohorts 1 and 2. Subsequently, implant resorption developed continuously for 1 year postoperatively.

**Figure 3 biomimetics-10-00101-f003:**
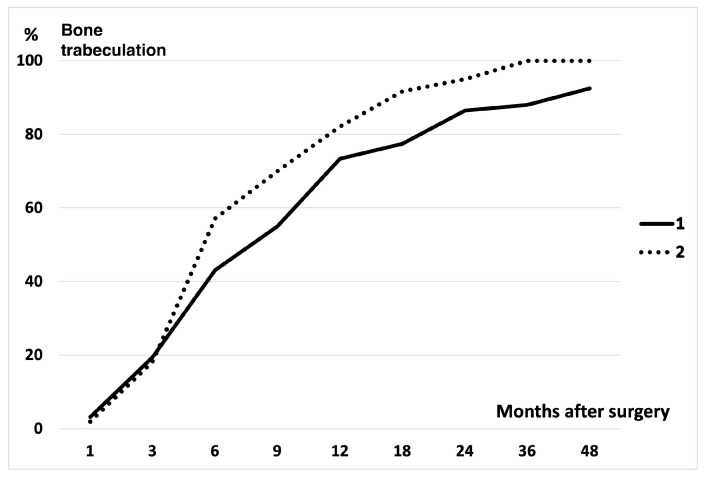
Bone trabeculation through the defect over time was observed in the comparison between cohorts 1 and 2. The rate of bone trabeculation increased significantly within 1 year postoperatively in cohorts 1 and 2. Bone trabeculation developed continuously for 1 year postoperatively.

**Table 1 biomimetics-10-00101-t001:** The demographic profile of 32 patients in Cohort 1.

Characteristics	Values or No. of Patients
Age (years)	
Mean	22.4 (6–70)
Gender	
Female	16
Male	16
Anatomical location	
Femur	9
Calcaneus	9
Tibia	5
Humerus	4
Patella	1
Talus	1
Radius	1
Ulna	1
Ilium	1
Site in long tubular bone	
Metaphysis or epiphysis only	17
Including diaphysis	3
Histology	
Simple bone cyst	12
Chondroblastoma	8
Fibrous dysplasia	4
Giant cell tumor	3
Aneurysmal bone cyst	3
Osteoid osteoma	1
Intraosseous ganglion	1
Tumor volume (cm^3^)	
Mean	14.4 (0.94–50.4)

**Table 2 biomimetics-10-00101-t002:** The demographic profile of 14 patients in Cohort 2.

Characteristics	Values or No. of Patients
Age (years)	
Mean	32.9 (10–45)
Gender	
Female	9
Male	5
Anatomical location	
Phalanges of finger	8
Metacarpal bones	5
Metatarsal bones	1
Histology	
Enchondroma	14
Tumor volume (cm^3^)	
Mean	1.2 (0.2–2.6)

**Table 3 biomimetics-10-00101-t003:** Clinical factors influencing radiographic assessment in Cohort 1.

	Radiographic Assessment			
	Implant Resorption (%)		Bone Trabeculation (%)	
	1 Year Postop.	2 Years Postop.	1 Year Postop.	2 Years Postop.
Sex				
Femal	82.8	91.7	75	89.6
Male	82.8	90.4	70.9	83.3
Age				
≤15 years	93.8 **	100 *	85.4 ***	96.4
>15 years	76.25 **	86.8 *	66.2 ***	82.4
Tumor size				
≤1 cm^3^	89.1	97.7	81.2 **	97.5 **
>1 cm^3^	76.6	85.7	65.6 **	78.6 **
Site in long tubular bone				
Including diaphysis	100	100	83.3	100
Metaphysis and/or epiphysis only	76.5	89.1	67.6	82.8

* *p* < 0.05, ** *p* < 0.03, *** *p* < 0.01.

**Table 4 biomimetics-10-00101-t004:** Clinical factors influencing radiographic assessment in Cohort 2.

	Radiographic Assessment			
	Implant Resorption (%)		Bone Trabeculation (%)	
	1 Year Postop.	2 Years Postop.	1 Year Postop.	2 Years Postop.
Sex				
Femal	88.9	99.4	80.6	92.9
Male	90	100	85	100
Age				
≤15 years	100	100	100	100
>15 years	87.5	97.2	79.2	94.4
Tumor size				
≤1 cm^3^	91.7	100	87.5	100
>1 cm^3^	87.5	97.5	78.1	90

## Data Availability

The original contributions presented in this study are included in the article. Further inquiries can be directed to the corresponding author.

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
