# Peer review of "Radiographic and Clinical Assessment of Unidirectional Porous Beta-Tricalcium Phosphate to Treat Benign Bone Tumors"

_biomimetics, 2025, doi:10.3390/biomimetics10020101_

Round 1
Reviewer 1 Report
Comments and Suggestions for Authors
In my humble opinion, this interesting and well-written manuscript might be published as is. No corrections are necessary. However, I am not a clinician and have never participated in clinical trials; therefore, an additional opinion from a clinician will be highly desirable.
Author Response
Comments: In my humble opinion, this interesting and well-written manuscript might be published as is. No corrections are necessary. However, I am not a clinician and have never participated in clinical trials; therefore, an additional opinion from a clinician will be highly desirable.
Responses: Thank you for your kind comments. We appreciate your review of our article. We have added and changed some sentences according to other reviewers’ comments.
Reviewer 2 Report
Comments and Suggestions for Authors
The article is suitable for publication, the only suggestion is that I would use a case-control group, with a group treated with unidirectional porous beta-tricalcium phosphate and others with the most common grafted substance at this time. It is not true that using a CT will significantly increase the radiation dose that the patient receives, if a low-dose CT was used, and in the extremities there is no risk of direct radiation to radiosensitive organs.
In Figure 1, we are provided with an initial study with CT and the follow-up after treatment with radiography. I would change the CT for an X-ray, so that they are comparable.
Author Response
Comments 1: The article is suitable for publication, the only suggestion is that I would use a case-control group, with a group treated with unidirectional porous beta-tricalcium phosphate and others with the most common grafted substance at this time.
Response 1: Thank you for bringing this to our attention. We agree with this comment. As we described in the Discussion as one of our limitations, I am afraid that since all patients received the same treatment during the study period, it was difficult to make a comparison with other types of bone filling materials or no filling. Therefore, we cannot use a case-control group as suggested by reviewer 2.
Comments 2: It is not true that using a CT will significantly increase the radiation dose that the patient receives, if a low-dose CT was used, and in the extremities there is no risk of direct radiation to radiosensitive organs.
Response 2: Thank you for bringing this to our attention. We agree with this comment. We conducted this study retrospectively and evaluated imaging that was routinely performed in outpatient clinics. The patients treated with bone grafting for bone tumors are usually followed up with plain radiography alone, and CT could be performed if something happens, such as fracture or tumor recurrence. Therefore, we wrote the following sentence in the Discussion.
"While CT may be able to more accurately demonstrate the incorporation of implanted UDPTCP and new bone formation for research purposes, repeated CT scans are not necessary for routine clinic after this type of surgery and may not be ethically justifiable due to the issue of radiation exposure, especially in young patients." (Lines 280-284)
Comments 3: In Figure 1, we are provided with an initial study with CT and the follow-up after treatment with radiography. I would change the CT for an X-ray, so that they are comparable.
Response 3: Thank you for bringing this to our attention. We agree with this comment. Therefore, we have changed the CT in Figure 1 to an X-ray (Figure 1a).
Reviewer 3 Report
Comments and Suggestions for Authors
The article "Radiographic and clinical assessment of unidirectional porous beta-tricalcium phosphate to treat benign bone tumors" (Toshiyuki Kunisada, Eiji Nakata, Tomohiro Fujiwara, Haruyoshi Katayama, Takuto Itano, Takanao Kurozumi, Teruhiko Ando, ​​Toshifumi Ozaki) is undoubtedly of interest from a clinical point of view for the effective restoration of bone cavities after removal of benign tumors of especially large sizes. However, there are issues in the work that require clarification and clarification.
1. The article materials often use "UDPTCP absorption". What do the authors mean by this concept? Perhaps they meant material resorption and the translation into English was incorrect.
2. It is necessary to label the axes in Figures 2 and 3.
Author Response
Comments 1: The article "Radiographic and clinical assessment of unidirectional porous beta-tricalcium phosphate to treat benign bone tumors" (Toshiyuki Kunisada, Eiji Nakata, Tomohiro Fujiwara, Haruyoshi Katayama, Takuto Itano, Takanao Kurozumi, Teruhiko Ando, ​​Toshifumi Ozaki) is undoubtedly of interest from a clinical point of view for the effective restoration of bone cavities after removal of benign tumors of especially large sizes. However, there are issues in the work that require clarification and clarification.
1. The article materials often use "UDPTCP absorption". What do the authors mean by this concept? Perhaps they meant material resorption and the translation into English was incorrect.
Response 1: Thank you for bringing this to our attention. We agree with this comment. Therefore, we have changed "absorption" to "resorption" throughout the text.
Comments 2: 2. It is necessary to label the axes in Figures 2 and 3.
Response 2: Thank you for bringing this to our attention. We agree with this comment. Therefore, we have included the axis titles in Figures 2 and 3. (Figures 2 and 3)